# Quantitative Evaluation of China’s Ecological Protection Compensation Policy Based on PMC Index Model

**DOI:** 10.3390/ijerph191610227

**Published:** 2022-08-17

**Authors:** Shengli Dai, Weimin Zhang, Linshan Lan

**Affiliations:** 1School of Public Administration, Central China Normal University, Wuhan 430079, China; 2School of Administration and Emergency Management, Jinan University, Guangzhou 510632, China

**Keywords:** ecological protection compensation policy, policy evaluation, PMC index model

## Abstract

As a comprehensive benefit coordination mechanism, ecological protection compensation has received increasing attention internationally. China has also introduced a series of policies to promote ecological protection compensation mechanism improvement. The evaluation of ecological protection compensation policies is the main basis for the formulation, adjustment and improvement of the policy. Based on the front-end perspective of policy formulation, this paper selects 10 ecological protection compensation policies issued by the State Council and various ministries in China from 2006 to 2021. A text mining-based PMC index evaluation system using the ROSTCM tool is constructed to quantitatively evaluate these 10 typical policies, and four representative policies are selected for comparative analysis. The overall design of China’s ecological protection compensation policies is reasonable, and 5 out of 10 policies have good PMC index scores, which are: P1, P2, P5, P7 and P9; and 5 are at acceptable levels, which are: P3, P4, P6, P8 and P10. However, the PMC surface and the concavity index indicate that there are still some limitations that need to be improved, including the obvious internal differentiation of the policies, the single effectiveness of the policies, and the insufficient incentive and guarantee measures. Finally, this paper provides suggestions for the optimization of ecological protection compensation policies accordingly.

## 1. Introduction

With the in-depth study of sustainable development and environmental–economic issues, the problem of inequality in the environmental field has been gradually incorporated into the research field. As a comprehensive mechanism, ecological compensation (equal to payment for ecosystem services, PES) can regulate the relationship between the beneficiaries and protectors of ecological protection, internalize ecological externalities through economic means [1,2,3,4] and, thus, promote environmental equity. Nowadays, it has increasingly become a popular policy tool, attracting great attention from international scholars. It is estimated that there are currently over 550 PES schemes with annual transactions of USD 36–42 billion by 2017 [5]. As a large flourishing developing country, ecological compensation in China is also a crucial issue in its efforts to balance economic development and ecological conservation and coordinate overall development and regional interests. Since the proposal of the 11th Five-Year Plan for National Economic and Social Development in 2005, the State Council has listed the construction of ecological protection compensation mechanisms as an annual work point. In 2010, ecological protection compensation was included in the B-class legislation plan [6]. Since then, the 18th Party Congress, the 3rd and 4th Plenary Sessions of the 18th Central Committee and the Opinions of the CPC (Central Committee and the State Council) on Accelerating the Construction of Ecological Civilization have taken the establishment of a sound ecological protection compensation mechanism as one of the important institutional guarantees for the construction of ecological civilization [7]. In 2016, the General Office of the State Council officially issued the Opinions on Sound Ecological Protection Compensation Mechanism, pointing out that the ecological protection compensation mechanism is essential for promoting the positive interaction between ecological protectors and beneficiaries, and to mobilize the whole society to protect the ecological environment [8]. It means that the top-level design of ecological protection compensation has made significant progress. The concept of “ecological compensation” has been officially changed to “ecological protection compensation”.

In accordance with the deployment of the Party Central Committee and the State Council, various regions and departments in China have been actively exploring ecological protection compensation in recent years and have gained valuable experience. In addition, the state is increasing the financial investment in ecological protection compensation year by year, covering forests, grasslands, mines, watersheds, soil and water conservation, key ecological function areas, oceans and other fields [7]. As of 2019, China has invested nearly 200 billion yuan in financial resources for ecological protection compensation, 15 provinces have participated in 10 cross-provincial watershed ecological compensation pilots, forest ecological efficiency compensation has achieved full coverage of national ecological public welfare forests and ecological compensation work has achieved bountiful results. However, at the same time, the related work is still confronted with difficulties such as the lack of standards for compensation, a single compensation method, and difficulty in mobilizing the enthusiasm of all parties.

Policy evaluation is an important and well-established part of the policy process, facilitating and feeding back to promote the ongoing effectiveness of policies that have been implemented or anticipating policies in the making [9], which plays an important role in public policy analysis [10]. As an important part of the system design of ecological civilization construction, ecological protection compensation policy has played a strong guiding and promoting role in the improvement in the ecological protection compensation mechanism. Therefore, to achieve the goal of ecological civilization system construction and solve the difficulties in the process of ecological protection, it is especially necessary to conduct evaluation on ecological protection compensation policies in the face of the deteriorating ecological environment. The existing studies have focused on the implementation effects of ecological protection compensation policies [11,12,13], and the evaluation methods have become more mature and developed into a comprehensive analysis method based on empirical evidence. However, the relevant research perspectives fail to evaluate ecological protection policies from the front-end perspective of the policy, the policy document itself. As an objective, accessible and traceable written record of the policy system and process, the policy document reflects the imprint of the government’s actions in dealing with public affairs [14]. A quantitative study of the policy itself based on the policy text itself helps to open the black box of the government’s decision-making process, thereby presenting public policy not as an “outcome” but as a “process” to the public [15]. Thus, it is of great significance to evaluate the policy text.

Therefore, based on ten policies issued at the national level from 2006 to 2021, this paper builds a PMC (Policy Modeling Consistency) index model [16] based on the text mining method to evaluate them quantitatively. This paper intends to address the following questions: What is the quality of current ecological protection policies? What are the problems and shortcomings in the design of the existing ecological protection compensation policies and how best to optimize the policy? Its contribution lies in constructing a PMC index model based on text mining, innovating the mode of policy evaluation and providing a theoretical basis and standard for future policy design, improvement and implementation. At the same time, its evaluation of ecological protection compensation policies reflects a good concern for environmental health equality issues. 

The remainder of this paper is organized as follows. Section 2 reviews the related literature. Section 3 introduces the research design. Section 4 presents the results and discussion. Section 5 is the conclusions.

## 2. Literature Review

### 2.1. Research on Ecological Protection Compensation

Ecological protection compensation is defined as an act of compensating ecological protectors by the beneficiary of ecological protection for cost expenditure and other related losses, in the manner of financial transfer payments or market transactions with money, materials or other non-material interests based on the comprehensive consideration of ecological protection cost, development opportunity cost and ecological service value [17]. Internationally, payment for ecosystem services (PES) is close to the concept of ecological protection compensation in China. It can be regarded as a voluntary transaction between ecosystem service users and ecosystem service providers, which generates corresponding ecosystem services based on the natural resource management regulations agreed by both parties and pays for them conditionally [18,19,20]. At present, scholars have mainly focused on the following three areas: the definition, mechanism construction, and evaluation and analysis of actual effects to provide feasible suggestions for policymakers. As to the definition, the theoretical basis of the scholars is mainly derived from Coase theorem [21,22], Pigou theory [2,23] or other perspectives. As to the eco-compensation mechanism construction, it includes three modes, namely, market transaction [24], government compensation [25] and non-market transaction [26]. At the same time, economic incentives and fairness are underlined in mechanism construction [21,27]. As to the practical benefits, scholars focus on the economy [28], ecology [29] and social benefits [30]. They try to evaluate the benefits through models [31] or cases [32] and dig up the driving factors [33]. In China, in the theoretical research of ecological compensation, scholars mainly focus on the concept of ecological compensation [34], policies and regulations [35,36], foreign experience [37,38], compensation type and standard [39,40], mechanism construction [7,41], etc., and show an interdisciplinary characteristic involving economic, management, legal and environmental fields, etc. In the case of policies and regulations, scholars have developed quantitative estimates of the environmental and economic benefits of ecological protection compensation policies [15], focusing on the actual impacts of ecological protection compensation policies [42,43,44].

### 2.2. Research on Policy Evaluation

Policy evaluation is a complex and systematic project aimed at measuring and evaluating policy items and providing a basis for policy formulation, adjustment and optimization [45]. Policy evaluation includes the evaluation of policy systems based on policy text, the evaluation of individual policy elements and evaluation of policy implementation effects [46]. Nowadays, the methods of policy evaluation have become increasingly diversified, with both qualitative studies based on cases and expert reviews and quantitative studies based on mathematical models. With the development of policy theory constantly being updated and deepened, the development of quantitative analysis methods and calculation software gradually matured, and the model was widely used. Most quantitative analysis methods try to establish the impact model of policy impact under certain assumptions, simulate the policy effect and evaluate the policy effect so as to make a “causal effect” evaluation. Thus far, a comprehensive evaluation method based on empirical analysis has been formed, such as text data mining [47,48], social network analysis [49], fuzzy comprehensive evaluation [50], etc. The commonly used empirical analysis tools are PSM-DID model analysis [51], tool variables [52] and synthesis control [53]. The evaluation content is diverse, involving the agricultural economy, finance, science and technology, medical treatment, environment and sustainable development.

The above studies have extensively discussed the concept definition, mechanism construction and practical influence of ecological protection compensation. The methods of policy evaluation have become more diversified and fruitful. However, there are the following areas that can be further improved: (1) As to the research object, although the existing studies have fully discussed the practical influence of ecological protection mechanisms, there is a lack of evaluation research on the quality of environmental protection policies text. (2) As to the research content, most of the existing studies are based on tracking the process aspects of policy implementation and enforcement. (3) At to the research direction, most existing studies analyze and evaluate relevant ecological protection compensation policy tools based on the policy end, i.e., policy effects, and less research quantitatively evaluates ecological protection compensation policies from the policy front-end, i.e., policy formulation perspective. Therefore, it is imperative to explore a new method to examine the quality of individual policies based on the policy making itself. The PMC index model was proposed by Ruiz Estrada [16], the Omnia Mobilis (everything is moving) hypothesis [54], in 2010. By establishing a multidimensional evaluation index system based on text mining, this method can effectively judge the consistency of policies, and reduce the subjectivity of evaluation and cost of policy analysis. In recent years, the application of the PMC index model has attracted great attention in academia and has become a prevailing method to evaluate the effectiveness of policies. International scholars have applied it to cultivated land protection policies [55], disaster relief policies [56], fishery insurance policy [57], long-term care insurance policy [58] and pork industry policy [59] evaluation. Zhang, a domestic scholar, applied it to regional science and technology innovation policy [60], financial policy [61], new energy vehicle subsidies [62], innovation policies of the state council [63], mass entrepreneurship and innovation [64] valuation, and laid the foundation of China’s policy evaluation. At present, the PMC evaluation model is mainly used in the research of the economy, science and technology and social security, but little attention has been paid to the ecological environment. 

These studies highlight that it is feasible and meaningful to use the PMC index model to evaluate China’s ecological protection compensation policy. Therefore, based on the above analysis, this paper uses the PMC index model from the perspective of text content analysis and selects the relevant policy texts of the ecological protection compensation policy from 2006 to 2021 for quantitative research. The research significance of this paper lies in focusing on the issue of ecological protection compensation under the background of environmental health equity, and applying the PMC index to policy evaluation. The purpose of this paper is to further enrich the literature in this field and provide a reference for policy optimization and innovation.

## 3. Research Design

### 3.1. Data Sources

By consulting policy documents and searching for keywords related to ecological protection compensation policy, this paper selects 10 important samples of ecological protection compensation analysis issued by the State Council and national ministries and commissions (Table 1). The time range is from 2006 to 2021. The year 2006 marks the second year of the establishment of the ecological protection compensation mechanism in China, and the concept was proposed in 2005 at the Fifth Plenary Session of the 16th CPC Central Committee. The year 2021 is the retrieval time for this study. The policy text of this paper is taken from webservice.pkulaw.cn to ensure its authority. This database is the earliest as well as the largest legal information service platform in China, containing all kinds of Chinese laws and regulations from 1949 to the present. As to the retrieval theme words, we selected “ecological protection compensation” as the keyword for policy filtering for its accuracy. As to retrieval strategy, the study contains two screening criterion: (1) The main body of the text is limited to the State Council and national ministries and commissions and does not involve government departments’ policies or industry standards. (2) Texts are directly related to the ecological protection compensation and currently valid. Finally, we obtained ten policy texts of high relevance as well as validity. 

### 3.2. Identification of Basic Policy Features 

In this paper, we used the data analysis software ROSTCM6.0 to word-sort the 10 ecological protection compensation policies collected, and ranked the word segmentation results in descending order of word frequency after the automatic recognition results were manually processed, i.e., the apparently useless words were eliminated. From Table 2, among the relevant policy texts to promote ecological protection compensation, it can be seen that the high-frequency words from the policy texts mainly show the following characteristics:

First, the policy theme is concentrated, and the focus is clear. Among the 30 keywords of national policy, the top five are: ecology, compensation, protection, mechanism and environment, according to the total frequency. These five key words account for 40.79% of the total frequency of the top 30 high-frequency keywords, showing that governments at all levels in China attach great importance to improving the compensation mechanism for ecological protection and making clear the key directions of the compensation mechanism for ecological protection. The formulation of compensation policies for ecological protection revolves around a relatively stable theme, and the themes are relatively concentrated. Secondly, the policy attaches importance to the improvement of the mechanism. From the key frequency table, it can be seen that “mechanism,” “system” and “governance” rank high among high-frequency words, indicating that they are the focus of policy attention. Furthermore, the policy attaches importance to the use of financial means to achieve the policy effect, and high-frequency words such as “fund,” “finance” and “payment” appear. With a background of ecological civilization construction, exploring the establishment of a diversified compensation mechanism for ecological protection requires the unification of powers and responsibilities and reasonable compensation. The government is good at using financial means to promote the balanced distribution of regional interests.

Network analysis and visualization appears to be an interesting tool to give the researcher the ability to see their data from a new angle. The semantics network is constructed based on the high-frequency words by data analysis software Gephi as shown in Figure 1. The closeness of each topic word is determined by the proximity of the nodes within the semantic network and the thickness of the connecting lines, and the thicker the line means the closer the relationship between two keywords. Meanwhile, the size of nodes represents the strength of centrality. If a node has more connections with other nodes, its degree centrality is stronger, which means the node is more important. The visualization results show that keywords “ecological protection”, “compensation mechanism”, “environment”, “Yangtze river basin” and “green development” have a high centrality. “Ecological protection”, “compensation mechanism” and “environment” are three main key words, which are located centrally, indicating that government have showed a high level concern towards environmental protection. Apart from this, such vocabularies as “money”, “institution improvement” and “transfer payment” are also an important concern, which indicates polices pay attention to the implementation of specific measures to help improve the ecological protection compensation mechanism.

### 3.3. Construction of PMC Index Model

In this paper, the PMC index model is used to quantitatively evaluate the ecological protection compensation policies issued by the state. The PMC index model is proposed by Estrada [16] based on the Omnia Mobilis hypothesis [54], which holds that everything in the world is moving and connected, and that all relevant variables should be considered as much as possible in the modeling and no relevant variables should be removed. The construction process consists of the following four steps: classification of variables and identification of parameters; creation of multi-input–output tables; measurement of PMC indices; and construction of PMC surfaces.

#### 3.3.1. Variable Classification and Parameter Identification

On the basis of summarizing the high-frequency words in ecological protection compensation text by ROSTCM6.0 and Kuang [55] and Dai’s [65] research, this paper constructed an index system of ecological protection compensation policy rating (Table 3). The index system consists of 10 first-level variables and 44 second-level variables, among which the first-level variables are: policy nature (X1), policy function (X2), policy timeliness (X3), policy field (X4), policy social benefits (X5), policy objects (X6), policy subjects (X7), policy incentive and restraints (X8), policy guarantee (X9) and policy disclosure (X10).

#### 3.3.2. Building a Multi-Input–Output Table

By constructing an alternative data analysis framework, the multi-input–output table can store a large amount of data to calculate any single variable. The establishment of a multi-input–output table is the basis for calculating 10 main variables, each of which includes n secondary variables, and there is no limit to the number of secondary variables, and the importance of secondary variables is the same, so it is not necessary to rank the importance. In order to give the same weight to the secondary variables, it is necessary to treat all variables in binary form here. Table 4 shows the specific results of the multi-input–output table established in this study.

#### 3.3.3. PMC Index Calculation

Based on ESTRADA’s research, this paper constructed a PMC index model of ecological protection compensation policy. The index model has four basic steps: (1)Construct variables according to the text of ecological protection compensation policy, including first-level and second-level variables.



(1)
X∼N[0,1]

(2)Create a multi-input–output table and base them on text mining and binary methods to assign specific values to secondary variables. Each parameter was coded to the binary values “0” or “1”.




(2)
X={X R:[0,1]} 

(3)Calculate the value of the first-level variable by Formula (3) in conjunction with the assignment of the second-level variable in the previous step. In the Formula (3), *t* is the ordinal number of first-level index, *j* is the ordinal number of second-level variable, *n* is the number of the second-level indexes.




(3)
$$X_{t}\left(\sum_{j=1}^{n}\frac{X_{tj}}{T\left(x_{tj}\right)}\right),t=1,2,3\cdots$$

(4)Sum up values of all variables by using Formula (4).




(4)
$$\begin{array}{c} \mathrm{PMC}=X_{1}\left(\sum_{a=1}^{9}\frac{X_{1a}}{9}\right)+X_{2}\left(\sum_{b=1}^{4}\frac{X_{2b}}{4}\right)+X_{3}\left(\sum_{c=1}^{3}\frac{X_{3c}}{3}\right)+X_{4}\left(\sum_{d=1}^{5}\frac{X_{4d}}{5}\right)+\\ X_{5}\left(\sum_{e=1}^{4}\frac{X_{5e}}{4}\right)+X_{6}\left(\sum_{f=1}^{4}\frac{X_{6f}}{4}\right)+X_{7}\left(\sum_{g=1}^{8}\frac{X_{7g}}{8}\right)+X_{8}\left(\sum_{h=1}^{6}\frac{X_{8h}}{6}\right)+X_{9}\left(\sum_{i=1}^{8}\frac{X_{9i}}{8}\right)+X_{10} \end{array}$$



According to Formula (4), the PMC index of 10 ecological protection compensation policies can be calculated and then rated. Based on existing research [55,65], the PMC index of environmental protection compensation policies could be divided into four levels of consistency (Table 5).

#### 3.3.4. PMC Surface Construction 

The PMC surface visualizes the PMC index in the form of a stereo image, which can visually display the advantages and disadvantages of the policy [16]. Considering the balance of the matrix, X10 is eliminated to form a third-order square matrix. Formula (5) is used for PMC surface calculation. In the PMC–Surface, the numbers 1, 2 and 3 mean the horizontal axis values of the matrix, and the series 1, 2 and 3 represent the ordinal values of the matrix. Different colored modules represent different scores of variables. The advantages and disadvantages of various policies can be judged according to the degree of depression on the surface. The convex part of the surface indicates a higher score on the corresponding evaluation variable, while the concave part indicates a lower score on the corresponding evaluation index.
(5)$$PMC-Surface=\left(\begin{array}{c} X_{1}\;X_{2}\;X_{3}\\ X_{4}\;X_{5}\;X_{6}\\ X_{7}\;X_{8}\;X_{9} \end{array}\right)$$

### 3.4. Research Question 

In this paper, a PMC index model based on text mining was constructed for policy quantitative evaluation. By analyzing PMC index score tables, and comparing PMC surface and radar charts of policies, this paper aims to answer following questions:(1)What is the quality of current ecological protection policies? By summing up and calculating the PMC index and classifying it hierarchically, we can understand the overall development level of ecological protection compensation policy.(2)What are the shortcomings in the overall design of the existing ecological protection compensation policies? The advantages and disadvantages of the various policies can be judged by the degree of concavity of the PMC surface. It will help to identify the shortcomings of the policy.(3)In what ways can future policies be improved? A comparative analysis of policies to clarify their strengths and weaknesses can help us better grasp the direction of policy improvement.

## 4. Results and Discussion

### 4.1. The PMC Index of 10 Samples

In this paper, the PMC index is calculated by establishing a multi-input–output table (the results are shown in Table 6). The scores of each of the 10 policies were derived and rated and ranked according to the PMC index calculation method, and the results are shown in Table 7. In order to present the degree of concavity and the advantages and disadvantages of the policies more visually, this paper draws a PMC surface of the 10 policies according to the PMC matrix (Figure 2, Figure 3, Figure 4, Figure 5, Figure 6, Figure 7, Figure 8, Figure 9, Figure 10 and Figure 11). According to the results, it could be found that:

The PMC index of the 10 ecological protection compensation policies fluctuates between 4 and 6 with an average score of 5.47, which indicates that the policy texts are categorized as acceptable as well as excellent. The ranking of the ten policies in order of score is: P2 > P7 > P1 > P5 > P9 > P4 > P3 > P8 > P10 > P6. Policy 2 is the opinion of the General Office of the State Council on the sound ecological protection compensation mechanism at the central level, which has a score of 6.31 and is rated as good, with a policy ranking of 1. Policy 6 is the notice of the Ministry of Finance on the issuance of transfer payments from the central government to local key ecological function areas, which has a rating of 4.21 and is at an acceptable level, with the lowest score among all policies. This indicates the existence of unevenness in the level of policies. According to the consistency criteria set in Table 5, we can classify the 10 policies into two categories for detailed discussion, the first category is the excellent level policies, they are: P1, P2, P5, P7 and P9; the second category is the acceptable type policies, they are: P3, P4, P6, P8 and P10. 

### 4.2. Specific Evaluation of Each Group of Ecological Protection Compensation Policies

#### 4.2.1. The “Good” Group of Policies

Policies P1, P2, P5, P7 and P9 are good policies, which shows that in recent years, the state has stepped up efforts to improve the construction of ecological civilization, accelerated the formation of an ecological protection compensation system in line with China’s national conditions and promoted the formation of green production methods and lifestyles. The above policy themes focus on the pilot establishment of compensation for watershed ecological protection and the improvement of horizontal ecological protection mechanisms, etc. The policy content is more scientific and comprehensive. Policy P1 is the opinions of the State Environmental Protection Administration to carry out the pilot work of ecological compensation. The policy involves economic, social, scientific and ecological aspects. It comprehensively uses various incentive and restraint mechanisms and provides a good guarantee for implementing the policy. Policy P2 is the opinion of the general office of the State Council at the central level on improving the compensation mechanism for ecological protection. This policy is rated as good and ranks first, which shows that this policy is well-considered in all dimensions. According to the release time, although the policy was released in 2016, the policy planning timeliness is long, and most of the later ecological protection compensation policies were released under the guidance of policy P2, which indicates that China has an excellent top-level design and complete overall planning in realizing ecological protection compensation. Policy P5 is a notice jointly issued by many departments on printing and distributing the action plan for establishing a market-oriented and diversified compensation mechanism for ecological protection. This policy puts forward some suggestions on the resource development compensation system, emission right allocation, water right allocation, carbon emission right offset mechanism, ecological industry and other aspects. It aims to improve the incentive mechanism and the investigation and monitoring system, strengthen technical support, and create good basic conditions for establishing a market-oriented and diversified ecological protection compensation mechanism. Policy P7 is a notice jointly issued by the Ministry of Finance and other departments to support and guide the pilot implementation plan of establishing a horizontal ecological compensation mechanism in the whole Yellow River basin. The policy aims to speed up the construction of the pattern of harnessing the upper, middle and lower reaches, the main river and tributaries, and the left and right banks, and promotes the excellent protection of the Yellow River in all provinces (regions) of the Yellow River Basin. The policy explores the improvement of the horizontal ecological compensation mechanism of the Yellow River Basin from four dimensions: platform construction, fund arrangement, mechanism construction and organizational guarantee. Policy P9 is the guiding opinion of the National Development and Reform Commission on accelerating the construction of the compensation mechanism for the ecological protection of Dongting Lake and Poyang Lake. This policy covers a wide range of fields, including the economy, society, science and technology, and ecology, and provides a guarantee for policy implementation from assessment, publicity and guidance, government supervision, social supervision, and scientific and technological innovation.

#### 4.2.2. The “Acceptable” Group of Policies

P3, P4, P6, P8 and P10 are acceptable policies, with scores of 4.99, 5.48, 4.20, 4.95 and 4.84, respectively, which are of an acceptable consistency, indicating that there is room for further improvement in the policies. The above policy subjects focus on the improvement and reform of the ecological compensation mechanism in river basins, and the improvement of the transfer payment system. On the whole, the policy is mainly deficient in terms of policy objects, policy subjects and the policy implementation guarantee. Policy 6 is the Notice of the Ministry of Finance on Printing and Distributing the Central Government’s Transfer Payment Measures to Local Key Ecological Function Zones. Its score is 4.20, which is acceptable, but it is at the bottom of the overall policy ranking. This policy aims to clarify the scope of transfer payment support and the principle of transfer fund allocation. However, it involves a single field, subject and recipient and fails to provide a good guarantee for policy implementation, thus the score is relatively low. Therefore, the policy can be improved from the aspects of X4, X6, X7 and X9. Policy P3 is the guidance issued by the Ministry of Finance and other departments on accelerating the establishment of a compensation mechanism for horizontal ecological protection upstream and downstream of the river basin. This policy score level is below average, and weak scores are mainly in X8 and X9. Therefore, the incentive constraints of the policy as well as the implementation guarantee will be the direction of future policy improvement. Policy P4 is the guidance of the Ministry of Finance on establishing and improving the long-term mechanism of ecological compensation and protection in the Yangtze River Economic Belt. The level of this policy score is approximately equal to the average, and the weak scores are mainly X6 and X9, i.e., a single policy receptor as well as insufficient implementation guarantees, which provides a direction for future policy improvement. Policy P8 is a notice jointly issued by the Ministry of Finance and other departments on printing and distributing the implementation plan to support the establishment of a horizontal ecological protection compensation mechanism in the whole Yangtze River basin. The level of this policy score is below average, with weak scores mainly for X6, X7 and X9, i.e., single policy objects and subjects as well as insufficient implementation guarantees. Policy P10 is an opinion on deepening the reform of the ecological protection compensation system jointly issued by the General Office of the CPC Central Committee and the General Office of the State Council, and the policy rating is acceptable. The level of this policy score is below average, deficient in X7, X8 and X9, providing paths for policy improvement. 

### 4.3. Comparative Analysis of Ecological Protection Compensation Policies

To evaluate the ecological protection compensation policy, we can compare the first-level variable score of the policy to be evaluated with the average score, or use the sag index to analyze the difference between the evaluation policy and the “perfect policy”, and find out the weak links in combination with the sag degree. This paper selects policies P2, P6, P7 and P10 for horizontal comparison by drawing radar chart as shown in Figure 12, and the scores are 6.30, 4.20, 6.10 and 4.84 in turn, among which policies P2 and P7 are good policies, and policies P6 and P10 are acceptable policies.
(1)X1 (policy nature). According to the regulations on the handling of official documents by Chinese party and government organs, policy nature consists of the number of documents issued and the type of policy documents, which could reveal the variability of different policies [66], and, in this paper, policy nature refers to the latter, including: the law, regulations, projects, opinions, notifications, decisions, plans and methods. The four policies are opinions or measures issued by the central government and its ministries, and the policy scores are balanced.(2)X2 (policy function). The policy function is an intrinsic property of the policy, referring to the role and effect that the policy can play in the process of implementation [67]. According to the PMC surface, the scores of policies P2 and P7 are 1, which indicates that the policy plays a good role in normative guidance, classified supervision, collaborative management and overall coordination, highlighting the effective governance of the government. In contrast, policies P6 and P10 scored 0.5, and there were no clear measures in X2.1 standard guidance, X2.3 collaborative management and X2.3 overall coordination, indicating that the policy should further detail its effects.(3)X3 (policy timeliness). The policy timeline refers to the time frame of the impact of the policy and reflects the overall design planning of the policy. The average value of the four policies is 0.33. Policy P2 is a long-term policy and policies P6, P7 and P10 are short-term policies, which shows that the development target time limit set by the content of the ecological protection compensation policy is relatively single, and the specific planning on long-term, medium-term and short-term development time limits is not comprehensive enough.(4)X4 (policy area). The policy area refers to the scope of policy influence, including economic, social, political, scientific and technological, and ecological. The scores of policies P2, P7 and P10 are all higher than the average score, which indicates that the policies cover a wide range of fields, including economy, society, science and technology and ecology, and the content is perfect. It is worth noting that policy 6 scored only 0.2 points in this item, only for the ecological field. From the content of the policy, it aims to promote the construction of ecological civilization and provide high-quality development and guide local governments to strengthen ecological, environmental protection by defining the scope of transfer payment support.(5)X5 (social benefits). The social benefits generated by the policy include environmental protection, sustainability, sound mechanism and win-win cooperation. The policy P7 score of X5 is 1, which is much higher than the average value, indicating that the policy is well-considered and pays attention to social benefits. The scores of policies P2, P6 and P10 are 0.50, 0.50 and 0.75, respectively, which are lower than the average value of 0.80. Therefore, future policy improvement can strengthen the attention to social benefits.(6)X6 (policy objects). Policy objects mainly refer to those who are affected by the policy and need to take corresponding measures [68]; it is the focus point of policy operation, and to a certain extent reflects the level and scope of the policy role. Policy P2’s score is 1. The policy receptor is the people’s governments of all provinces, autonomous regions and municipalities directly under the Central Government, ministries and commissions of the State Council and institutions. In China, government departments at different administrative levels have played an essential role in improving the compensation mechanism for ecological protection. The higher the administrative level of the institution, the higher the administrative level, and the stronger the coordination ability when implementing the ecological protection compensation policy, so policy P2 has a better policy effect in this item. Policies P6, P7 and P10 scored 0.25 in this item, and further attention can be paid to the cooperation of policy objects, that is, executive departments, in future policy improvement.(7)X7 (policy subjects). The policy subject includes a variety of subject forms such as single subject and multi-subject association, which is the beginning of the policy operation. Generally speaking, the administrative level and management scope of policy-issuing agencies directly affect the efficiency of policy implementation [69].The score of P7 policy is 0.50, which is higher than the average, while the scores of P2, P6 and P10 are all 0.13, which is lower than the average. It can be seen that P7 has a good policy effect in this item.(8)X8 (policy incentive and restraints). Policy incentives and constraints are measures to motivate the subjects to actively implement their responsibilities to promote the effective implementation of policies in the process of policy implementation. The scores of P2, P6 and P10 on X8 are higher than the average, which indicates that the incentive constraint is perfect. For P7, the policy score is only 0.33, which is lower than the average. Therefore, more attention should be paid to implementing incentive and restraint measures in future policy improvement.(9)X9 (policy guarantee). In order to facilitate the smooth implementation of the policy, a series of guarantees are usually put in place when the policy is released [70]. According to the surface chart, policy 2 is higher than the average value in the X9 execution guarantee office, which indicates that the policy pays attention to the execution guarantee work to promote the implementation of the policy. The scores of P7 and P10 are 0.50, which is at the average level. The policy score for P6 is only 0.25, which is also the reason for the low overall score of the policy. Therefore, in the future, the improvement of the policy can further pay more attention to the implementation of safeguard measures.

## 5. Conclusions

In this paper, a combination of the text mining method and the PMC index model was used to evaluate the consistency of ecological protection compensation policies since 2006 in China. The characteristics of 10 selected policies were further discussed. The main conclusions are given below:

By constructing the PMC index evaluation index system, it could be found that the overall design of China’s ecological protection compensation policy is relatively reasonable. Among the ten ecological protection compensation policies, five national policies score good consistency, and five policies score acceptable consistency. This shows that against the background of ecological civilization construction, our government has attached great importance to the protection of the ecological environment, and strives to give play to the guiding role of ecological protection compensation policy to promote the improvement of the ecological protection mechanism and regional coordinated and sustainable development.

By drawing the PMC index model, we can understand the advantages and disadvantages of target policies from a three-dimensional perspective, and trace the secondary indicators to determine the path of policy optimization. According to the results, it could be found that there is still some room for improvement in China’s ecological protection policy. First of all, the overall score of the policy is not perfect, and the internal differentiation of the policy is evident. Overall, the policy scores remained acceptable and above, but failed to reach the perfect consistency. The main points of the 10 ecological protection compensation policies are policy nature, policy timeliness, policy objects, policy subjects, policy incentives and constraints, and policy implementation guarantees. There is a significant difference in policy scores, with policy P6 scoring the lowest, 4.20 points, and policy P2 scoring the highest, 6.30 points. According to the PMC surface chart, the degree of depression is P6—P10—P8—P3—P4—P9—P5—P1—P5—P7—P2 from strong to weak, and the weak links of each policy are different. Secondly, the main objects and subjects of policy are singular, and inter-ministerial coordination needs to be strengthened. The average scores for the 10 policies in policy objects and policy subjects are 0.25 and 0.33, which are below the overall average score. As to policy subjects, only 4 out of 10 policies are jointly issued by many departments. As to policy objects, the implementing agencies of the 10 policies are mostly specific departments or government agencies, enterprises and institutions. In addition, the policy timeliness is mainly based on short-term and medium-term planning. The long-term planning score is low. Finally, the incentive and restraint measures need to be further improved. The scores of the 10 ecological protection compensation industrial policies are low in talent incentives, special funds, tax incentives, etc. The areas covered by the policies need to be further enriched. According to the research results, suggestions for the improvement of China’s ecological protection compensation policy are as follows:

First, improve policy functions and enrich the nature of policies. Ecological protection is a systematic project involving many aspects to strengthen its guiding role. It is necessary to give full play to the role of ecological protection compensation policies in standardized guidance, classified supervision, coordinated guidance and overall coordination to achieve good social benefits. Because of the particularity of ecological protection compensation policy, governments at all levels should focus on improving the incentive and restraint mechanism, give full play to the financial investment to promote the smooth development of regional ecological compensation and improve the efficiency of fund use. Second, strengthen departmental cooperation. Policy improvement should strengthen overall coordination, promote the establishment of inter-ministerial coordination mechanisms composed of finance, environmental protection, development and reform, water resources, science and technology, transportation, forestry and other departments, and study and solve major problems in the construction of ecological protection compensation mechanism in time. Each department should guide the ecological protection compensation work in this field to provide more standardized and comprehensive policy support. Third, focus on top-level design and long-term planning. Establishing a sound compensation policy for ecological protection is a long-term task. It is suggested that future policies should coordinate short-term goals and long-term planning, continue to increase the transfer payment compensation for key ecological functional areas and gradually improve their basic public service level.

Taking China’s ecological protection compensation as the research object, this paper combines text mining with the PMC index model, constructs a PMC index model of ecological protection compensation policy and makes a quantitative evaluation of ecological protection compensation. To wrap up, its innovation lies in three aspects: First, in terms of research content, this paper focuses on the topic of ecological protection compensation in the context of environmental health equity, i.e., socio-economically disadvantaged areas tend to sustain enhanced ecological degradation, hence are in need of additional compensation, and applies the PMC index model to policy evaluation. The findings are conducive to a comprehensive understanding of the characteristics of ecological protection compensation policies and can enrich the research in this field. Second, in terms of research perspective, this paper evaluates ecological protection compensation policy from the front part of the policy, which is conducive to tracking the whole process of governmental decision making and further enriching the research in the field of policy evaluation. Third, in terms of research methods, this paper integrates text mining and a PMC index model to construct a combined qualitative and quantitative method, which is conducive to breaking through the defects of a single method. However, there are also some limitations:

First, the sample selection is not comprehensive enough. Although this paper selects 10 representative policy texts, it is difficult to avoid ignoring or omitting other related policies. On the one hand, the research sample is only selected from national level ecological protection compensation policies, and does not cover the local level. On the other hand, there is a lack of analysis of specific policies for forests, grasslands, wetlands, deserts, oceans, etc. Second, the PMC index evaluation model also has some limitations, despite its advantages of operability and low cost. First, the setting of PMC indexes needs to be optimized. The setting of it emphasizes universality, and the coverage and expansion of the index need to be further clarified to make it more relevant. Second, the calculation method of the PMC index is a simple summation and averaging, without integrating the information of each policy evaluation index, and thus cannot measure the relationship between different evaluation indexes of policies. Thirdly, manual annotation is used for policy text annotation, and there is a certain subjectivity. Lastly, the policy text mining analysis is an ex ante evaluation and does not continue to track the concrete implementation of the policy to dig up its actual effect on promoting the balance of regional ecological protection. Therefore, the following aspects can be considered in the future policy improvement: First, the future policy can further increase the sample capacity and increase the policy investigation for local policies and specific areas. At the same time, the evaluation of other specialized policies can be increased so that policy suggestions can be more scientific. Secondly, non-standard variables can be preset, and specific problems can be analyzed to improve the pertinence of the PMC index model. In addition, in the future, we can consider combining the BP neural network method to capture the nonlinear relationship with the PMC index model for policy modeling so as to improve the scientificity of the quantitative evaluation.

## Figures and Tables

**Figure 1 ijerph-19-10227-f001:**
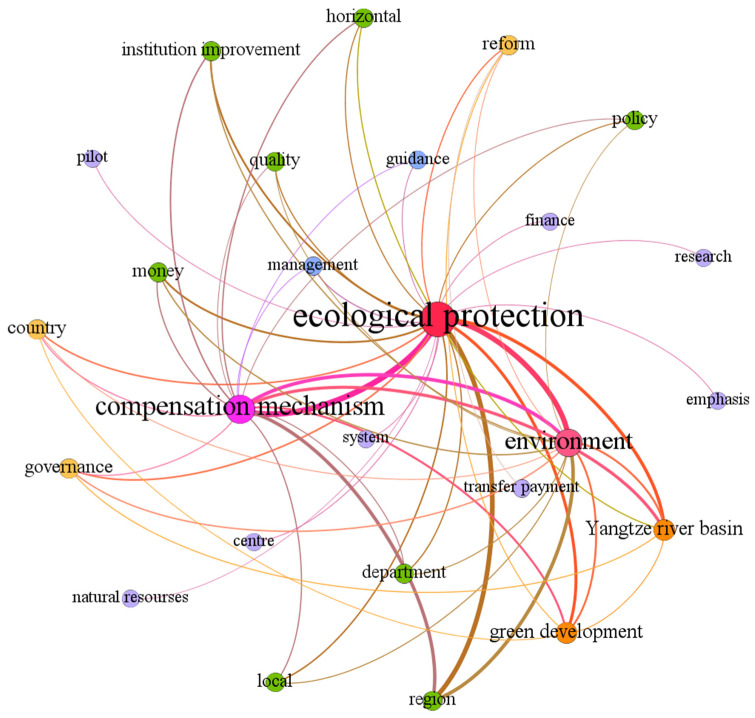
Keyword Network Map.

**Figure 2 ijerph-19-10227-f002:**
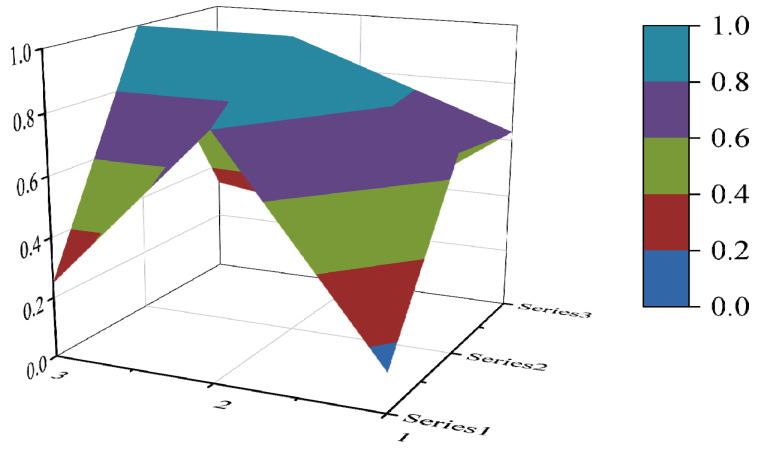
The PMC surface of P1.

**Figure 3 ijerph-19-10227-f003:**
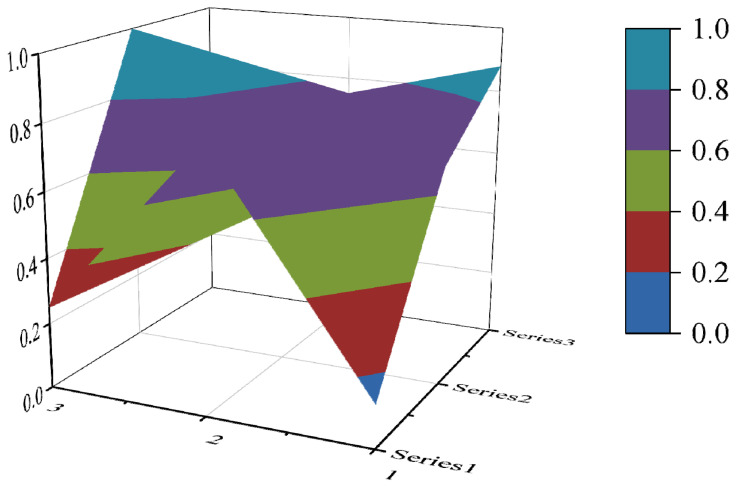
The PMC surface of P2.

**Figure 4 ijerph-19-10227-f004:**
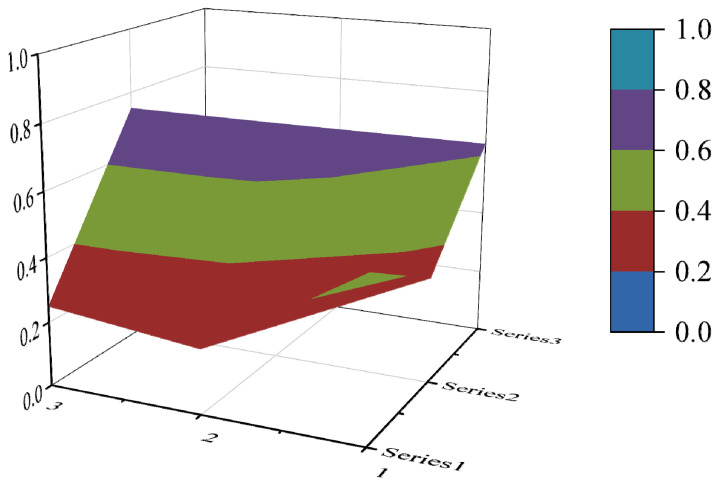
The PMC surface of P3.

**Figure 5 ijerph-19-10227-f005:**
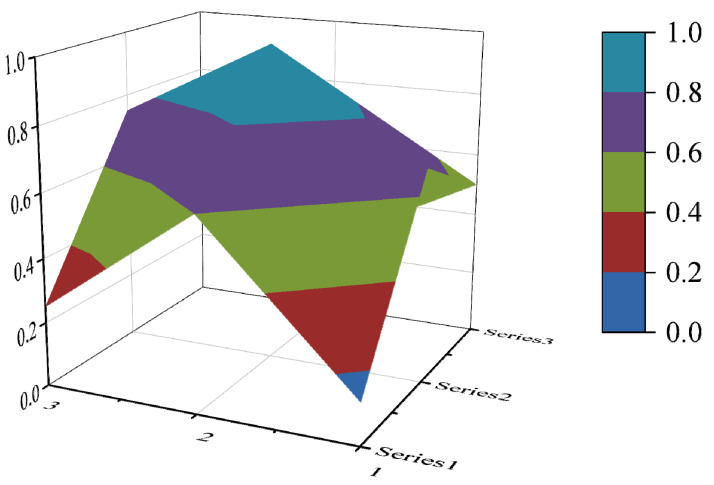
The PMC surface of P4.

**Figure 6 ijerph-19-10227-f006:**
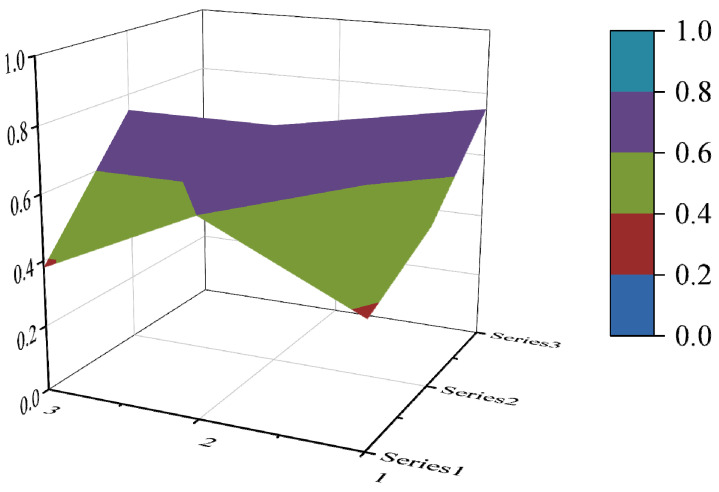
The PMC surface of P5.

**Figure 7 ijerph-19-10227-f007:**
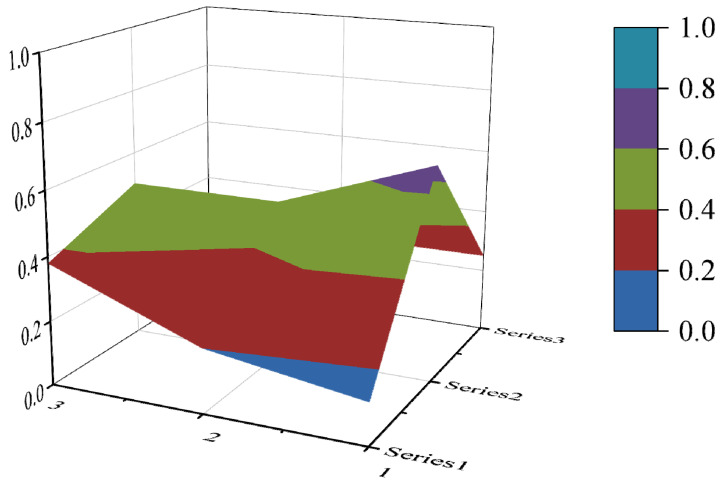
The PMC surface of P6.

**Figure 8 ijerph-19-10227-f008:**
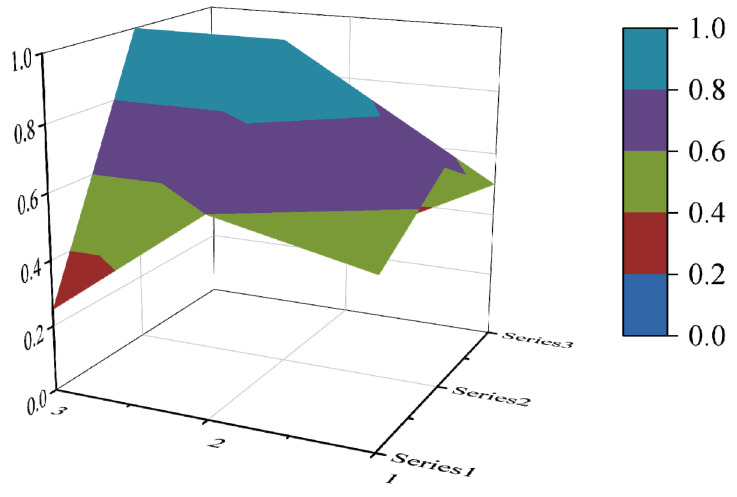
The PMC surface of P7.

**Figure 9 ijerph-19-10227-f009:**
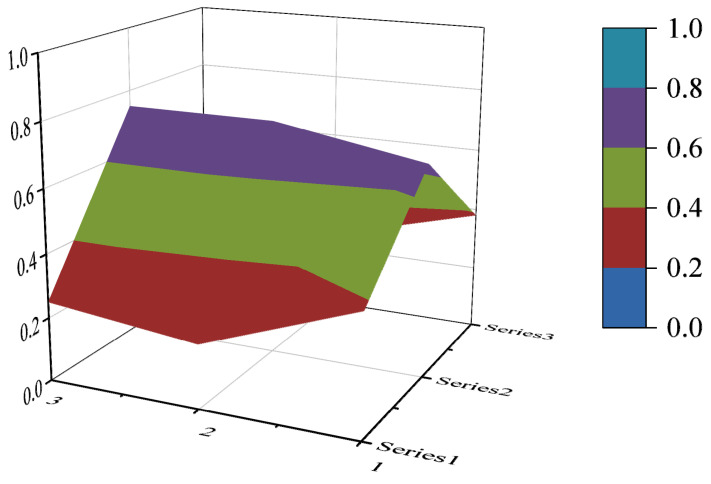
The PMC surface of P8.

**Figure 10 ijerph-19-10227-f010:**
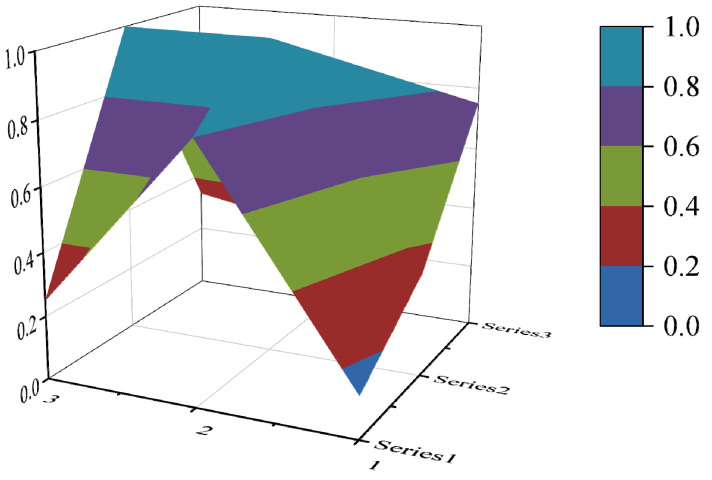
The PMC surface of P9.

**Figure 11 ijerph-19-10227-f011:**
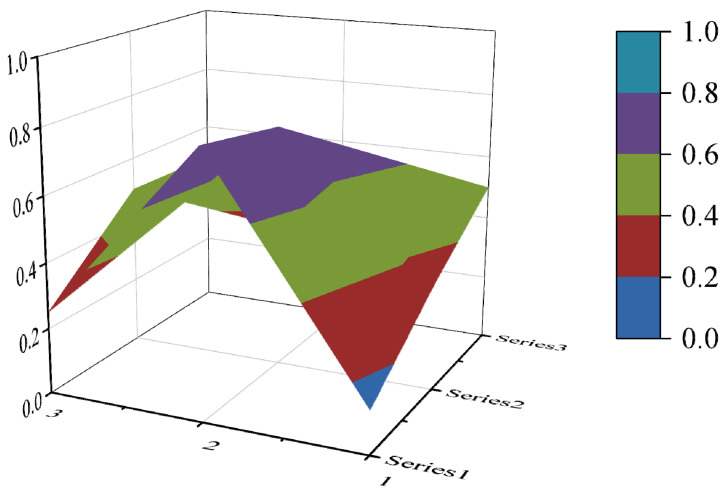
The PMC surface of P10.

**Figure 12 ijerph-19-10227-f012:**
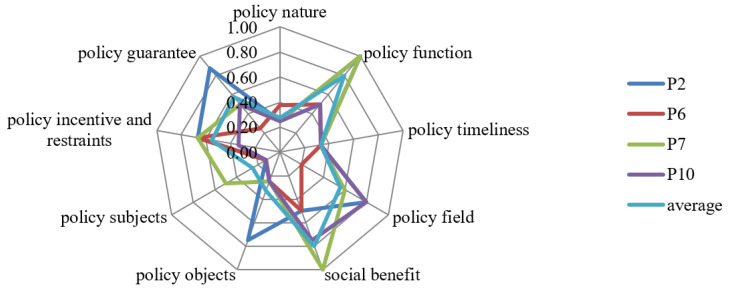
Radar chart comparison of P2, P6, P7 and P10.

**Table 1 ijerph-19-10227-t001:** Sample policies for PMC index model.

Code	Policy Name	Date Issued
P1	Guidance on the development of ecological compensation pilot work	24 August 2007
P2	Opinions on the sound ecological protection compensation mechanism	28 April 2016
P3	Guidance on accelerating the establishment of horizontal ecological protection compensation mechanisms upstream and downstream of the watershed	20 December 2016
P4	Guidance on the establishment of a sound long-term mechanism for ecological compensation and protection of the Yangtze River Economic Belt	13 February 2018
P5	Notice on the issuance of “the establishment of market-oriented, diversified ecological protection compensation mechanism action plan”	28 December 2018
P6	Notice on the issuance of the “support to guide the Yellow River basin-wide pilot implementation plan for the establishment of horizontal ecological compensation mechanism”	9 May 2019
P7	Notice on the issuance of the “support to guide the Yellow River basin-wide pilot implementation plan for the establishment of horizontal ecological compensation mechanism”	20 April 2020
P8	Notice on the issuance of the Implementation Plan to Support the Establishment of Horizontal Ecological Protection Compensation Mechanism in the Whole Yangtze River Basin	16 April 2021
P9	Guiding Opinions on Accelerating the Construction of Ecological Protection and Compensation Mechanism of Dongting Lake and Poyang Lake	28 May 2021
P10	Opinions on Deepening the Reform of Ecological Protection Compensation System	12 September 2021

**Table 2 ijerph-19-10227-t002:** Policy text key words’ frequency statistics table.

Serial Number	Vocabulary	Frequency	Serial Number	Vocabulary	Frequency
1	Ecology	910	16	Pilot	84
2	Compensation	524	17	Governance	81
3	Protection	516	18	Local	79
4	Mechanism	300	19	Yangtze River	74
5	Environment	255	20	Green	70
6	Basin	235	21	Transfer	67
7	Development	144	22	Reform	64
8	Money	128	23	Finance	64
9	Construction	126	24	Payment	62
10	Region	121	25	Resource	61
11	Country	111	26	Policy	59
12	Institution	98	27	System	59
13	Department	95	28	Ministry of Finance	58
14	Horizontal	89	29	Nature	48
15	Emphasis	85	30	Functional Zone	48

**Table 3 ijerph-19-10227-t003:** PMC evaluation index system and evaluation standard.

Level Indicator	The Secondary Indicators	Define
X1 Policy nature	X1.1 law	Whether the policy belongs to laws; if yes, it is 1, if no, it is 0.
	X1.2 regulation	Whether the policy belongs to regulations; if yes, it is 1, if no, it is 0.
	X1.3 project	Whether the policy belongs to project; if yes, it is 1; if no, it is 0.
	X1.4 opinion	Whether the policy belongs to opinion; if yes, it is 1, and if no, it is 0.
	X1.5 notification	Whether the policy belongs to notification; if yes, it is 1, if no, it is 0.
	X1.6 decision	Whether the policy belongs to decision; if yes, it is 1; if no, it is 0.
	X1.7 plan	Whether the policy belongs to the plan; if yes, it is 1; if no, it is 0.
	X1.8 method	Whether the policy belongs to the method; yes is 1, no is 0.
X2 Policy function	X2.1 Normative guidance	Whether the policy involves the function of normative guidance; if yes, it is 1, and if no, it is 0.
	X2.2 Classified oversight	Whether the policy involves the function of classified supervision; if yes, it is 1; if no, it is 0.
	X2.3 Collaborative management	Whether the policy function involves the function of collaborative management; if yes, it is 1; if no, it is 0.
	X2.4 Overall coordination	Whether the policy function involves the function of overall coordination; if yes, it is 1; if no, it is 0.
X3 Policy timeliness	X3.1 short-term	Whether the policy prescription is less than 3 years; if yes, it is 1, if no, it is 0.
	X3.2 Medium-term	Whether the policy prescription is 3–5 years; if yes, it is 1, if no, it is 0.
	X3.3 Long term	Whether the policy prescription is greater than 5 years; if yes, it is 1, if no, it is 0.
X4 Policy field	X4.1 Economy	Whether the policy involves the economic field; if yes, it is 1, if no, it is 0.
	X4.2 Society	Whether the policy affects social life; if yes, it is 1, if no, it is 0.
	X4.3 Politics	Whether the policy involves the political field; if yes, it is 1, if no, it is 0.
	X4.4 Science and technology	Whether the policy involves science and technology field; if yes, it is 1, if no, it is 0.
	X4.5 Ecology	Whether the policy has environmental implications; if yes, it is 1, if no, it is 0.
X5 Policy social benefits	X5.1 Environmental protection	Whether the policy contributes to environment protection; if yes, it is 1; if no, it is 0.
	X5.2 Sustainability	Whether the policy contributes to sustainable development; if yes, it is 1; if no, it is 0.
	X5.3 Sound mechanism	Whether the policy produces has the utility of a sound mechanism; if yes, it is 1; if no, it is 0.
	X5.4 Win-win cooperation	Whether the policy contributes to win-win cooperation; if yes, it is 1; if no, it is 0.
X6 Policy objects	X6.1 National ministries and commissions	Whether the policy object is a national ministry; if yes, it is 1, and if no, it is 0.
	X6.2 Local government	Whether the policy object is a local government; if yes, it is 1, if no, it is 0.
	X6.3 Enterprise and public institution	Whether the policy object is an enterprise or a public institution; if yes, it is 1; if no, it is 0.
	X6.4 Other	Whether the policy object is another relative department; it is 1, if no, it is 0.
X7 Policy subjects	X7.1 Ministry of finance	Whether the policy subject is the Ministry of Finance; if yes, it is 1; if no, it is 0.
	X7.2 Ministry of Environmental Protection	Whether the policy subject is the Ministry of Environmental Protection; if yes, it is 1; if no, it is 0.
	X7.3 National Development and Reform Commission	Whether the policy subject is the National Development and Reform Commission; if yes, it is 1, if no, it is 0.
	X7.4 Ministry of Water Resources	Whether the policy subject is the Ministry of Water Resources; if yes, it is 1; if no, it is 0.
	X7.5 Ministry of Science and Technology	Whether the policy subject is the Ministry of Science and Technology; if yes, it is 1; if no, it is 0.
	X7.6 Ministry of Transport	Whether the policy subject is the Ministry of Transport; if yes, it is 1; if no, it is 0.
	X7.7 Forestry Bureau	Whether the policy subject is the Forestry Bureau; if yes, it is 1, and if no, it is 0.
	X7.8 other related departments	Whether the policy subject is other related departments; if yes, it is 1; if no, it is 0.
X8 Policy incentive and restraints	X8.1 Economic incentive	Whether the policy includes measures for economic incentives; if yes, it is 1, and if no, it is 0.
	X8.2 Tax benefits	Whether the policy includes measures for tax benefits; if yes, it is 1, and if no, it is 0.
	X8.3 Financial subsidy	Whether the policy includes measures for financial subsidy; if yes, it is 1, and if no, it is 0.
	X8.4 Convenient service	Whether the policy includes measures for convenient service; if yes, it is 1, and if no, it is 0.
	X8.5 Administrative penalty	Whether the policy includes measures for administrative punishment; if yes, it is 1, and if no, it is 0.
	X8.6 Capital investment	Whether the policy includes measures for the capital investment; if yes, it is 1, and if no, it is 0.
X9 Policy guarantee	X9.1 Assessment	Whether the policy involves assessment; if yes, it is 1; if no, it is 0.
	X9.2 Publicity and guidance	Whether the policy involves publicity and guidance; if yes, it is 1, and if no, it is 0.
	X9.3 Self-regulation	Whether the policy involves industry self-regulation; if yes, it is 1; if no, it is 0.
	X9.4 Government regulation	Whether the policy involves government regulation; if yes, it is 1, and if no, it is 0.
	X9.5 Law rules	Whether the policy involves legal rules; if yes, it is 1, and if no, it is 0.
	X9.6 Policy support	Whether the policy involves policy support; if yes, it is 1; if no, it is 0.
	X9.7 Social supervision	Whether the policy involves social supervision; if yes, it is 1, and if no, it is 0.
	X9.8 Technological innovation	Whether the policy involves technological innovation; if yes, it is 1, and if no, it is 0.
X10 policy disclosure		Whether the policy is open and transparent; if yes, it is 0, if no, it is 1.

**Table 4 ijerph-19-10227-t004:** Multi-input–output table.

X1	X2
X1.1, X1.2, X1.3, X1.4, X1.5, X1.6, X1.7, X1.8	X2.1, X2.2, X2.3, X2.4
X3	X4
X3.1, X3.2, X.3.3	X4.1, X4.2, X4.3, X4.4, X4.5
X5	X6
X5.1, X5.2, X5.3, X5.4	X6.1, X6.2, X6.3, X6.4
X7	X8
X7.1, X7.2, X7.3, X7.4, X7.5, X7.6, X7.7, X7.8	X8.1, X8.2, X8.3, X8.4, X8.5, X8.6
X9	X10
X9.1, X9.2, X9.3, X9.4, X9.5, X9.6, X9.7, X9.8	X10

**Table 5 ijerph-19-10227-t005:** Consistency Categories.

PMC Index	0~3.968	3.969~5.548	5.549~7.127	7.128~7.920
Evaluation	Low consistency	Acceptable consistency	Good consistency	Perfect consistency

**Table 6 ijerph-19-10227-t006:** Input–output policy table.

	X1						X2				X3			
	X1.1	X1.2	X1.3	X1.4	X1.5		X2.1	X2.2	X2.3	X2.4	X3.1	X3.2	X3.3	
P1	0	0	0	1	0	…	1	0	0	1	0	0	1	
P2	0	0	0	1	0		1	1	0	0	0	0	1	
P3	0	0	0	1	0		1	0	1	0	0	0	1	
…							…							
P9	0	0	0	1	0		1	0	1	1	0	0	0	
P10	0	0	0	1	0		0	1	0	1	0	0	0	
	…						…							
	X8						X9							
P1	0	0	0	0	…	1	0	1	0	1	1	1	…	0
P2	1	0	0	0		1	1	1	0	1	1	1		1
P3	0	0	1	0		0	1	1	0	1	0	0		0
P9	0	0	0	0		0	1	1	0	1	0	1		1
P10	0	0	1	0		0	1	0	0	1	1	1		0

**Table 7 ijerph-19-10227-t007:** Policy PMC index and type.

	X1	X2	X3	X4	X5	X6	X7	X8	X9	X10	PMC Index	Type	Rank
P1	0.25	1.00	0.33	0.80	1.00	0.25	0.13	0.67	0.63	1.00	6.06	Good	3
P2	0.25	1.00	0.33	0.80	0.50	0.75	0.13	0.67	0.88	1.00	6.31	Good	1
P3	0.25	0.75	0.33	0.20	0.75	0.25	0.50	0.33	0.63	1.00	4.99	Acceptable	7
P4	0.25	0.75	0.33	0.60	1.00	0.25	0.13	0.67	0.50	1.00	5.48	Acceptable	6
P5	0.38	0.75	0.33	0.60	0.75	0.50	0.38	0.50	0.75	1.00	5.94	Good	4
P6	0.38	0.50	0.33	0.20	0.50	0.25	0.13	0.67	0.25	1.00	4.21	Acceptable	10
P7	0.25	1.00	0.33	0.60	1.00	0.25	0.50	0.67	0.50	1.00	6.10	Good	2
P8	0.25	0.75	0.33	0.20	0.75	0.25	0.38	0.67	0.38	1.00	4.96	Acceptable	8
P9	0.25	1.00	0.33	0.80	1.00	0.25	0.13	0.33	0.75	1.00	5.84	Good	5
P10	0.25	0.50	0.33	0.80	0.75	0.25	0.13	0.33	0.50	1.00	4.84	Acceptable	9
Total	2.76	8.00	3.30	5.60	8.00	3.25	2.54	5.51	5.77	10.00	54.73	-	-
Average	0.28	0.80	0.33	0.56	0.80	0.33	0.25	0.55	0.58	1.00	5.47	-	-

## Data Availability

The datasets used or analyzed during the current study are available from the corresponding author on reasonable request.

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
