# Peer review of "Quantitative Evaluation of China’s Ecological Protection Compensation Policy Based on PMC Index Model"

_ijerph, 2022, doi:10.3390/ijerph191610227_

Round 1
Reviewer 1 Report
(1)The overall frame is unclear. (2)There is plenty of room for improvement in the rigor of text description.(3)The analysis of the study findings lacks pertinence.(4)The innovation of the PMC has not been effectively reflected. (5)The abstract needs to be rewritten and should complement the methods adopted, quantified results, main conclusions and key findings.(6)The Introduction requires major modifications. It is necessary to supplement the research progress in the fields, the findings of the author's literature analysis, leading to the scientific problems that the author expects to address. (7)There are still many problems in the formula, calculation and analysis.
Author Response
On behalf of all the co-authors, I would like to take this great opportunity to appreciate reviewer#1 for the careful and valuable comments concerning our paper entitled Quantitative Evaluation of China’s Ecological Protection Compensation Policy-Based on PMC Index Model (Manuscript No.: ijerph-1796198). The summary of our work as written by this reviewer#1 is precise and constructive, do great help for us to better sharpen our paper. Here we have addressed all the comments carefully and the revised portions are highlighted within the document in red-color. Point-by-point responses to the nice reviewer#1 are listed in the attached document. We hope that the addressing of these comments would improve the quality of this manuscript.

Reviewer 2 Report
1. Although the abbreviation PMC is obvious from the authors' point of view, Introduction needs a brief explanation. This explanation appears in line 2 on page 4, whereas in the previous text the abbreviation PMC appears, e. g. in the last paragraph of Introduction.
2. I suggest subsection 3. 3. 5 with a research question and, if necessary, a research hypothesis which can be briefly examined in the chapter “Conclusion and enlightenment. ”
3. Figure 8 has S. . . instead of Series 1 / 2 / 3. Can this be improved?
4. Minor corrections to the literature – some journals are written in italics, others are not.
Author Response
On behalf of all the co-authors, I would like to express our sincere appreciation to reviewer#2 for your insightful and valuable comments for our paper entitled Quantitative Evaluation of China’s Ecological Protection Compensation Policy-Based on PMC Index Model (Manuscript No.: ijerph-1796198). We value your acknowledgement and constructive suggestions of this manuscript. Thus, we have revised the manuscript according to the comments and highlighted modified part within the document in red-color. Point-by-point responses to the nice reviewer#2 are listed in the attached document. We hope that our responses and edits are satisfactory.

Reviewer 3 Report
This is a very interesting paper concerning important and difficult issue: evaluation of the state/region ecological policy. Documents of the State/region ecological policies are relatively new tools of sustainable management in many countries. In addition, these dokuments are very complex (involve many aspects) and multidisciplinary. So, the evaluation of effectiveness of these documents (policy) is really difficult the more that the experience of countries/scientists in this regard is not large. So, I estimate highly this paper which presents original, partly new, methodical approach to evaluation of ecological polices in the country/region. This is a new contribution in methodology in this field.
The paper concerns an important problem - the multi-aspect evalualtion of the state environmental protection compensation policy, case study: China. The Authors have presented a special and original metodical approach, how to evaluate effectiveness of a such policy and give ways for improvement - how to design the next better version of ecological policy.
Detailed notes:
Section 1. INTRODUCTION. It is good. Authors formulated aims of the reserarch and gave good scientific background as a study justification and indication of a gap in research carried out so far.
Section 2. LITERATURE review - is OK.
Section 3. MATERIALS AND METHODS. This section includes the most valuable element of this paper: original, partly new, methodological approach to evaluation of the State ecological policy - quantitative evaluation as regards policy text, individual elements building a policy and implementation of the policy (effectiveness of the policy) using mathematical models (e.g. PMC surface Index), case studies nad original data.
Authors have taken under research selected 10 ecological protection compensation policies in China (from years 2006 - 2020) = 10 important samples of ecological policies. What are the criteria of this selection? (suggestion to supplement it by Authors).
Authors carefully explained their methodological approach (not easy one) by series of tables and figures with comments - showing all stages of this methodical approach. It is very good and valuable from the point of methodology. E.g. Table 1 shows important and interesting types of ecological compensation mechanisms/systems (from different regions of China) and Table 2 shows the set of key words selected by Authors for needs of policy text evaluation.
Section 4. EMPIRICAL ANALYSIS OF ECOLOGICAL PROTECTION COMPENSATION POLICY EVALUATION. In fact, there are research results in this section - with a poor discussion (there is no a section with discussion in the whole paper). My suggestion is: give the topic to section 4: RESULTS AND DISCUSSION - ? Subsections are OK. Of course, there is a need to develop better elements of discussion (now there is too few citations). Content of the section 4 is very interesting, including way of application of the original methodical approach presented in Section 3.
By the way: editorial error in the caption under Fig. 12: two times "of" is.
Section 5. CONCLUSION AND ENLIGHTENMENT. This section is well done and very interesting
My suggestion is: change topic to: CONCLUSIONS.
In general: Presented methodical approach of evaluation of the state ecological policy has a lot of advantages which are emphasied by Authors. However, the Authors have not indicated disadvantages of this approach. So, my suggestion is to supplement the weaknesses of this method - in CONCLUSIONS and in the section 4. E.g. is this methodical approach relatively easy to apply by the state/region authorities?
Editorial error: lack of a space between "since'" and "2015" (page 16, the first paragraph of the section 5).
GENERAL OPINION. Summerizing all, I estimate the paper highly. It is worth publishing. It is interesting for readers (scientists and decision-makers). The original methodological approach is the most valuable - in the field of knowledge and methodology. Its contribution in knowledge mainly consists on: construction of the PMC index model based on policy text mining, innovation of the way of ecological policy evaluation, providing of more theoretical basis and standards for future policy design, improvement and implementation.
Author Response
On behalf of all the co-authors, I would like to express our sincere appreciation to reviewer#3 for your insightful and valuable comments for our paper entitled Quantitative Evaluation of China’s Ecological Protection Compensation Policy-Based on PMC Index Model (Manuscript No.: ijerph-1796198). We greatly value your acknowledgement and constructive suggestions of this manuscript. Thus, we have revised the manuscript according to the comments and highlighted modified part within the document in red-color. Point-by-point responses to the nice reviewer#3 are listed in the attached document. We hope that our responses and edits will be satisfactory.
